# OpenReview forum: "Implicit degree bias in the link prediction task"
_ICML.cc/2025/Conference — ICML 2025 poster_

### Official Review · Reviewer_C2zd · 2025-02-27

**Overall Recommendation:** 4

**Summary:**

This paper studies the degree bias when benchmarking link prediction methods. Since most graphs follow a power-law distribution, the connected edges are very likely to be formed by two nodes with high degrees. However, the negative edges are often sampled by node, which results in a set of edges formed by nodes with lower degrees. This can cause an unfair benchmark such that LP methods only favoring node degree can achieve much better performance compared to others capturing predictive structural features like common neighbors, shortest-paths. Then the paper proposes a degree-corrected benchmark method and shows that it can provide a more robust evaluation benchmark for LP tasks.

**Claims And Evidence:**

Yes. The claims made in the submission are clear and supported by evidence.

**Essential References Not Discussed:**

NA

**Experimental Designs Or Analyses:**

- It would be better to list all the graph datasets and their statistics in the appendix. It would be easier for readers to know what datasets are included in the experiments.

**Methods And Evaluation Criteria:**

Yes.

**Other Comments Or Suggestions:**

Again, it would be better to list all the graph datasets and their statistics in the appendix. It would be easier for readers to know what datasets are included in the experiments.

**Other Strengths And Weaknesses:**

Strength:
- The study of LP method benchmarking is meaningful, which can encourage to focus on LP methods that are better at capturing nuanced graph structures rather than naive node degrees.

- The analysis of the paper is easy to understand. The writing is clear.

- The empirical results support the assumptions and claims in the paper.

Weakness:
- The authors can discuss more about the practical use cases where degree-corrected benchmark aligns, beyond just recommendation task.

- Two OGB datasets, DDI and PPA, can also be included in the discussion. These two datasets are highly dense and have high average node degree. It would be interesting to know whether the proposed benchmark can also reflect the nature of these graphs well.

**Questions For Authors:**

NA

**Relation To Broader Scientific Literature:**

- It suggests a more fair way to evaluate LP methods, which may encourage the development of modern LP methods to have a more practical impact on real-world problems.

**Theoretical Claims:**

Didn't check the correctness of the proofs. The theoretical claims are straightforward to understand.

---

> ### Author Rebuttal · Authors · 2025-03-31
>
> Thank you for your thoughtful review and constructive feedback! We also appreciate your remarks on the clarity of our analysis and writing. We have addressed each point raised as follow.
>
> ## Dataset statistics
> We agree that including all graph dataset statistics improves clarity. We have added a summary table in Appendix A listing the number of nodes, edges, average degree, and density for each dataset used in our experiments.
>
> ## DDI and PPI networks
> We would like to clarify that both the DDI and PPI networks from the OGB benchmark were already included in the original submission. We have revised the manuscript to make this more explicit by providing the tables.
>
> ## Broader practical use cases
> Thank you for raising this point. While we do not make new claims about specific downstream applications in this version, we have a sentence in the discussion section noting that the broader applicability of degree-corrected edge sampling. As mentioned, given the widespread use of edge sampling across graph machine learning tasks, we believe our findings have implications that extend beyond LP evaluation, potentially informing the design of benchmarks and training protocols more broadly.
>
> Thank you again for taking the time to review our work!

---

### Official Review · Reviewer_2ZUx · 2025-03-14

**Overall Recommendation:** 3

**Summary:**

Observing existing link prediction models often sample partial negative edges for evaluation; this paper hypothesized that this sampled evaluation includes degree bias and would cause the link predictor model to over-fit to capture node degree signal in making a prediction. After empirical and theoretical analysis, this paper successfully demonstrates the degree of bias, proposes a degree unbiased negative sampling method, and demonstrates that the newly proposed benchmark would result in different rankings for some link prediction models and better align with recommender systems.

**Claims And Evidence:**

1. One concern here is the investigated bias is discovered based on the basic negative sampling strategy. However, many recent works have come up with more advanced negative sampling strategy and therefore, I am not sure whether the discovered bias would appear in other places, such as [1]

[1] Li, Juanhui, et al. "Evaluating graph neural networks for link prediction: Current pitfalls and new benchmarking." Advances in Neural Information Processing Systems 36 (2023): 3853-3866.

**Essential References Not Discussed:**

This paper has comprehensively reviewed existing works addressing link prediction bias.

**Experimental Designs Or Analyses:**

1. Although their described setup was once the most widely used, more recent works have proposed many different link prediction negative sampling strategies, and I am not very sure whether the discovered negative bias would also exist in these new methods.

2. Throughout the analysis, the paper does not leverage any advanced machine learning model for link prediction, such as the more recently proposed BUDDY [1] and NCN [2]. Since both explicitly model the structures into the link prediction decision-making, it will be interesting to see how their performance relates to the degree distribution.

[1] Chamberlain, Benjamin Paul, et al. "Graph neural networks for link prediction with subgraph sketching." arXiv preprint arXiv:2209.15486 (2022).

[2] Wang, Xiyuan, Haotong Yang, and Muhan Zhang. "Neural common neighbor with completion for link prediction." arXiv preprint arXiv:2302.00890 (2023).

**Methods And Evaluation Criteria:**

I have checked the designed method, especially the experiment, to discover the link prediction bias and can confirm it makes sense to me.

**Other Comments Or Suggestions:**

See the questions below.

**Other Strengths And Weaknesses:**

Strengths:
(1) This paper investigates a widely used technique for evaluating link prediction performance. The sampling bias discovered in this technique has never been systematically investigated before.

(2) This paper provides a rigorous justification, not only in terms of theoretical analysis (e.g., derived the relationship between the degree distribution and the PA link prediction AUCROC) but also empirically analyzed the performance.

(3) this paper also demonstrates several implications of using degree-corrected benchmarks, one for aligning with recommendation tasks (which is more aligned with real-world applications) and one for learning community structure.

Weakness:
(1) Throughout the analysis, the paper does not leverage any advanced machine learning model for link prediction, such as the more recently proposed BUDDY [1] and NCN [2]. Since both explicitly model the structures into the link prediction decision-making, it will be interesting to see how their performance relates to the degree distribution.

[1] Chamberlain, Benjamin Paul, et al. "Graph neural networks for link prediction with subgraph sketching." arXiv preprint arXiv:2209.15486 (2022). [2] Wang, Xiyuan, Haotong Yang, and Muhan Zhang. "Neural common neighbor with completion for link prediction." arXiv preprint arXiv:2302.00890 (2023).

(2) The motivation of the section 3.3 is unclear. If it aims to show that the proposed benchmark captures a lower node degree, would it be better to directly demonstrate that the learned node embedding can lead to better degree prediction performance? If it is to show the benchmark capture more salient graph structures, I wonder if there is any application where the link prediction performance requires capturing the substructures.

**Questions For Authors:**

(1) In Figure D, the author attributes the advantages of the PA model over others to its explicit ability to capture degree signals. However, I’m curious about models that outperform PA. What are these models, and do they also capture degree signals? It might be better to analyze those better than PA and derive insights on whether they can capture degree signals.

(2) What about testing in the inductive setting, where the growth of the network might deviate from its expected behavior?

**Relation To Broader Scientific Literature:**

This paper discloses the link prediction bias in a new degree-related way.

Although the link prediction itself is a very long-standing problem, the discovered degree bias is a fresh perspective and may inform several implications related to link prediction, such as recommender systems where many negative sampling techniques are used.

**Theoretical Claims:**

I have checked the theoretical claim.

---

> ### Author Rebuttal · Authors · 2025-03-31
>
> First of all, thank you for your time and for providing constructive comments on our manuscript!
>
> ### > many recent works have come up with more advanced negative sampling strategy, and therefore, I am not sure whether the discovered bias would appear in other places, such as [1] Li, Juanhui, et al. "Evaluating graph neural networks for link prediction: Current pitfalls and new benchmarking." Advances in Neural Information Processing Systems 36 (2023): 3853-3866.
>
> HeaRT benchmark is in fact precisely the benchmark proposed by the paper suggested by the reviewer.
> We identified that the HeaRT benchmark in fact deviates from the retrieval tasks, and, when used as an unsupervised training objective, undermines the learning of communitiy structure, even if the communities are well separated.
> From theoretical standpoint, our method is more advanced, as it is grounded on the theoretical understanding of the degree bias, which is different from the HeaRT benchmark, which is a heuristic based on empirical observation of different bias.
> From empirical standpoint, the simplicity is the strength of our method, as it is computationally cheap, easy to implement. And being simple does not mean that it is less effective than more complex ones; in fact, we have demonstrated that it is more effective in terms of the alignment with retrieval tasks, and learning of community structure, than the HeaRT benchmark.
>
> > ### The paper does not test advanced ML models (e.g., BUDDY, NCN). How do these models relate to degree distribution and the proposed benchmark?
>
> We have in fact included BUDDY in our experiments in our previous version of the manuscript. We emphasize that the degree bias is not specific to specific models, but a fundamental issue that is present in the training data. Since it is not agnostic to the model, the degree bias affects all models equally.
>
> > ### What is the goal of Sec 3.3? Is it about degree prediction or structure modeling? Clarify the intended contribution and application relevance.
>
> We appreciate the request for clarification. The goal of Section 3.3 is to demonstrate that correcting degree bias enables models to capture more meaningful graph structure beyond node degrees.
> This matters because link prediction is often used as a convenient unsupervised training objective for learning node representations.
> We use community structure as a test case because it reflects higher-order organization not reducible to degree. The results show that models trained under our benchmark recover community assignments more accurately, highlighting the benchmark's value for representation learning.
>
> > ### Figure D discusses PA's advantage due to degree, but what about models that outperform PA? Do they also capture degree?
>
> Our intention is not to claim that PA is the top-performing model. We use PA as a controlled baseline because it relies solely on degree, making it ideal for isolating the effect of degree bias. It is expected that methods leveraging additional structural signals, e.g., distance and subgraph similarity, can outperform PA. However, we observe that many such "advanced" models fail to do so consistently, especially under standard benchmarks (Figure 2a). This is precisely the issue we aim to highlight.
>
> >###  How does degree bias behave under inductive link prediction where network growth may deviate from expected behavior?
>
> We thank the reviewer for this question. We have, in fact, discussed this point in our discussion. To clarify, we explain this point in detail below.
>
> In transductive settings, link prediction occurs between nodes in the training graph, while inductive settings involve predicting links for new nodes or unseen graphs. Inductive link prediction often uses similar approaches to transductive ones, such as classifying edges as positive or negative or retrieving the top-k most likely edges.
> Importantly, the degree bias persists in the inductive setting. For example, when sampling edges from new nodes uniformly at random, a node with $k$ new edges is $k$ times more likely to be selected than a node with $1$ new edge, mirroring the degree bias in the transductive setting.
>
> Transductive link prediction is as equally important as inductive ones and should not be neglected; for example, in citation networks, predicting missing citations between existing papers (transductive) is just as crucial as predicting citations for newly published papers (inductive), as it helps discover missing knowledge connections, assess research impact accurately, and maintain citation quality control.

---

### Official Review · Reviewer_9kyj · 2025-03-16

**Overall Recommendation:** 3

**Summary:**

This paper argues that the existing benchmark for link prediction inevitably involves implicit degree bias during the positive and negative edges sampling since both sampling distributions are theoretically proven to be inconsistent. Consequentially, existing link prediction methods implicitly overvalue the characteristic of degree, in particular for graphs with high heterogeneity. To solve this issue, the degree-corrected link prediction benchmark is proposed, which samples negative edges with a similar distribution as the positives. Extensive experiments and analysis show that the proposed benchmark facilitates the capability of capturing comprehensive graph structures (i.e., community structure) instead of node degree simply and boosts strong baselines (e.g., GAT) as a result.

**Claims And Evidence:**

One of the significant claims, that the proposed degree-corrected benchmark aligns better with recommendation tasks, is not convincing as follows:
1.	Alignment is only evaluated between AUC-ROC and VCMPR, the classification ability of recommenders, while neglecting the ranking ability (i.e., NDCG), another significant metric for recommenders.
2.	The algorithms used for evaluations seem only to be link prediction methods rather than specific graph recommendation methods.
3.	The graphs used for evaluations are also unknown, which may not be the typical user-item bipartite graph as well. Whether the proposed benchmark works on bipartite as well as its evidence (e.g., relation between bipartite and unipartite, analyses on the sparsity, etc) remains unclear.
To sum up, the evaluation setting for this claim is not convincing to argue that the proposed benchmark can be reliable in recommendation tasks/settings. Whereas, this claim would rather show an alignment and consistency between two evaluation metrics using the proposed benchmark.

**Essential References Not Discussed:**

To the best of my knowledge, most of the essential references have been discussed, despite those related to the questions below.

**Experimental Designs Or Analyses:**

All experimental designs and analyses have been carefully checked.

**Methods And Evaluation Criteria:**

1.	The motivation of the proposed benchmark is insufficient. First, the authors argue that implicit degree bias exists in positive sampling and straightforwardly force the negative sampling to adhere to the same distribution without explanation. It seems to be adding extra bias among negative edges and ignoring the existing bias among positive edges. Second, two more heuristic benchmarks based on the proposed one are neglected: one is to force the positive sampling to adhere to the uniform distribution, and the other is to trade off two distinct distributions by fixing one node on the positive edge as an anchor and uniformly sampling another node to form the negative edge (e.g., pairwise BPR[1]).
2.	Regarding the proposed negative sampling benchmark, it seems to implement the data augmentation on the negative edge candidate sets, duplicating those negative edges by the degree of end nodes. It serves as the hard negative sampling since high-degree nodes that used to form positive edges are more likely to form negative edges than before. However, no related work or discussion on data augmentation and hard negative sampling were discussed in this paper. Moreover, the methods used for evaluations are outdated. Therefore, it is doubtful whether the proposed benchmark is crucial for learning comprehensive graph structure and gaining substantial ranking changes for the latest link prediction methods compared with other benchmarks.

[1]: Rendle, S., Freudenthaler, C., Gantner, Z., & Schmidt-Thieme, L. (2009). BPR: Bayesian Personalized Ranking from Implicit Feedback. UAI

**Other Comments Or Suggestions:**

1.	Section 4, paragraph 2, “To better [understand] the contribution…”
2.	Appendix B, equation 10, “\Phi(z^-)=\int^{z^-}_{[-]\infty}…”
3.	Appendix D.5, equation 17, “…\frac{…}{[min](C,m_i)}”
4.	Appendix D.5, equation 18, “RBO([U_{k,1},U_{k,2}],p):=…”

**Other Strengths And Weaknesses:**

Strengths:
1.	This paper discloses a critical issue in the link prediction task. The analysis and solutions should have a broad influence on the certain domain and pave the way for future studies.
2.	The fundamental proofs are firm and reasonable.
3.	This paper also clearly points out the limitations for further research.

Weaknesses:
The presentation and architecture of this paper need to be improved:
1.	Subfigures should be carefully cropped and put in an appropriate position to ensure clarity and illustration.
2.	The order of the different sections in the appendix needs to be reorganized to retain contextual coherence.
3.	Section 4 (Discussion) can be split into different subsections to make it more readable.

**Questions For Authors:**

1.	In Appendix D.8., extra experimental results for the LFR benchmark, varying the average degree <k>, community size, and degree, should be shown in Figures 11 and 12. How do you tell different results on the figure by the certain variable (e.g., <k>=25/50)?
2.	Open question: Is it possible that degree bias is one of the key features for graph learning and how to manipulate it properly matters indeed?

**Relation To Broader Scientific Literature:**

While this paper refers to broader scientific literature on the domain of link prediction and sampling bias, the key contributions of this paper, which disclose the implicit degree bias and propose a corresponding new benchmark as a solution, should be innovative.

**Theoretical Claims:**

All proofs for theoretical claims appear to be correct.

---

> ### Author Rebuttal · Authors · 2025-03-31
>
> Thank you for your thorough and meticulous review and many constructive comments!
>
> > ### Alignment is only evaluated between AUC-ROC and VCMPR, ...while neglecting the ranking ability (i.e., NDCG)
>
> We agree with the point and now use NDCG@k as our evaluation metric and report VCMPR@C results in Appendix. The findings are consistent, i.e., degree-correction yields 44% graphs above RBO >0.6, for HeaRT and original yield 1% and 27% of graphs above RBO >0.6, respectively.
>
> > ### The algorithms used for evaluations seem only to be link prediction methods rather than specific graph recommendation methods.
>
> Link prediction and recommendation are closely intertwined, and the distinction between them is not clear-cut. Both tasks rely on similar modeling techniques. For example, BPR trains recommendation models using triplets (user, positive item, negative item). BPR models learn ranking from pairwise comparisons of positive edges and negative edges, which is effectively the link prediction task we focused on. OGB datasets also follow a retrieval-style classification evaluation (e.g., ogbl-collab and ogbl-ddi). Thus, we believe that evaluating link prediction models in retrieval-like settings, as we do, is appropriate.
>
> > ### Whether the proposed benchmark works on bipartite ... remains unclear.
>
> All graphs in our experiments are unipartite. This choice aligns with existing benchmarks such as ogbl-citation2 and ogbl-ddi, and with prior work on link prediction [Li et al. 2024a, Huang et al. 2023, Mao et al. 2023, Menand & Seshadhri 2024]. We agree that assessing benchmark behavior in bipartite graphs is important. However, our goal is to establish and validate the benchmark in the unipartite setting as a first stepping stone on this issue. To clarify this point, we added new text to mention this future work.
>
> > ### It seems to be adding extra bias among negative edges and ignoring the existing bias among positive edges.
>
> In link prediction, positive edges reflect the observed reality of the graph. They are the ground truth and must remain unchanged. Negative edges, by contrast, are synthetic. They are created for training and evaluation, and their distribution is not fixed.
>
> An intuitive analogy is a clinical trial. Positive edges are like actual outcomes observed in patients who received the treatment. We shouldn't bias these empirical evidence. Negative edges, on the other hand, are like control groups, which are constructed to serve as a comparison baseline. These can be designed in various ways (e.g., random, matched by age or condition), depending on the study goals.
>
> If the control group is systematically different---say, composed of less healthy individuals---the evaluation of the treatment will be skewed. Likewise, in link prediction, if negative edges are sampled without considering node degree, they form an unfair comparison group (Appendix D), leading to benchmarks that reward models for exploiting degree imbalance rather than learning meaningful graph structure.
>
> > ### two more heuristic benchmarks ...  pairwise BPR[1].
>
> We have extended our analysis to include BPR-style asymmetric sampling (Appendix C). While this changes the sampling distribution, we confirmed that it does not eliminate the influence of node degree on evaluation metrics such as AUC-ROC.
>
> > ### no related work or discussion on data augmentation and hard negative sampling
>
> We respectfully disagree with labeling our method as data augmentation. Unlike augmentation,  our method cannot generate new positives, nor does it expand the dataset; its goal is not diversity, but distributional alignment for fair evaluation and learning.
>
> We clarify how our method differs from hard negative mining. Hard negatives are typically model-dependent: they are selected dynamically based on how difficult they are for a given model to classify (see Zhang & Stratos, ACL 2021; Xuan et al., ECCV 2020). In contrast, the degree bias we address is model-agnostic. It arises from the statistical distributional mismatch between the positive and negative edges before training.
>
> > ### How do you tell different results on the figure by the certain variable (e.g., <k>=25/50)?
>
> Each curve is not a function of the variable but shows the results for a method in a different graph configuration. Figs. 3, 13, 14 together show the consistency of the performance gains across different settings.
>
> > ### Open question
>
> Degree is indeed a valuable signal. Our goal is not to discard degree, but to prevent its overemphasis caused by biased negative sampling, which inflates the performance of degree-based methods like PA.
>
> biokg_drug exemplified this point, which has a strong rich-club structure—94% of edges connect the top 10% highest-degree nodes. Even with degree correction, PA still achieves an AUC-ROC of 0.9, confirming that degree remains predictive when appropriate.
>
> Thanks again for your thoughtful and constructive comments which have improved our manuscript!

---

### Official Review · Reviewer_vU2g · 2025-03-17

**Overall Recommendation:** 3

**Summary:**

This paper is focused on the link prediction task and it shows how the sampling procedure applied in the evaluation of link prediction methods is biased towards high degree nodes. More specifically, the selection of random negative pairs to be distinguished against positive pairs leads to negative pairs connecting low degree nodes. This issue is analyzed both empirically and theoretically. To address the issue, the paper proposes a degree-correlated sampling procedure that generated negative pairs that have similar degrees as the positive ones. Using this new benchmark, they show that preferential attachment achieves higher performance than predicted using the standard sampling procedure.

**Claims And Evidence:**

In section 3.3, the claims around Hits@K are kinda misleading. In recommendation systems, usually the whole workflow is done in a couple steps, essentially retrieval and ranking, where retrieval systems are usually nearest neighbor-based (any LP method with dot-product decoder) and ranking systems are usually pairwise methods (e.g., subgraph LP method such as SEAL). And the authors claim around Hits score is sort of putting a retrieval metric under the ranking, as in a nearest neighbor-based system, it's quite intuitive to calculate the similarity of 1 user against all items and get the top K, and that's also where metrics such as Recall@K and Hits@K are used. On the other hand, like the author said, it's hard for ranking methods to calculate the similarity between all pairs due to the high complexity. But in reality, they never need to, as they usually have a much smaller candidate set, i.e., the top K output by the retrieval system.

Considering that the recommendation task is the most relevant real world application to the task of LP, I'd suggest the authors to re-work on section 3.2, and also corresponding evaluations.

**Essential References Not Discussed:**

n/a

**Experimental Designs Or Analyses:**

see my comment above

**Methods And Evaluation Criteria:**

there's no proposed method

**Other Comments Or Suggestions:**

n/a

**Other Strengths And Weaknesses:**

s1. Having a good set of benchmark is very critical for the healthy development of the field, and the authors pointed out some fetal problems of existing benchmarks.

s2. The main hypothesis of the paper is supported by empirical and theoretical results.

**Questions For Authors:**

n/a

**Relation To Broader Scientific Literature:**

n/a

**Theoretical Claims:**

seems correct to me

---

> ### Author Rebuttal · Authors · 2025-03-31
>
> We thank the reviewer for their time and for pointing out the important distinction between retrieval and ranking in recommendation systems!
>
>  > ### Response to "Claim and Evidence" section
>
> In response to the reviewer's comment along with the comment from reviewer 9kyj, we have now used NDCG@K as our main evaluation metric for the retrieval task. The findings are consistent, i.e., degree-correction yields 44% graphs above RBO >0.6, for HeaRT and original yield 1% and 27% of graphs above RBO >0.6, respectively, which is in line with our previous result for VCMPR@C.
>
> Additionally, we have reworked the paragraph to make clear the two-step retrieval pipeline suggested by the reviewer as follows:
>
> - Recommendation systems typically involve two steps: retrieval and ranking. First, the retriever selects a smaller candidate set from the entire node set, after which the ranker orders these candidates. In our experiments, we adopt a two-stage retrieval pipeline to reflect this practice. Initially, a retriever selects the top candidate neighbors per node using its similarity function. Then, a ranking model ranks these candidates. Both the retriever and the ranker are based on the same similarity function for the embedding- and topology-based models, but for pairwise link prediction models (i.e., BUDDY and MLP), we use the local random walk (LRW) to retrieve the candidate sets because enumerating all node pairs is computationally challenging. We chose LRW because it is among the best performing methods in the retrieval task.
>
> This makes clear that we are using a two-step retrieval pipeline, which wasn't clear form the previous version of the manuscript.
>
> > ### In section 3.3, the claims around Hits@K are kinda misleading
>
> We thank the reviewer for raising this important point. Let us clarify the intent of our original statement regarding Hits@K:
>
> -  Text in the previous version of the manuscript: *"In the recommendation task, directly optimizing recommendation metrics such as Hits@K requires ranking all possible node pairs for each node, which is computationally infeasible for large networks"*.
>
> We understand that this sentence may have been interpreted as referring to inference, particularly in the context of a typical two-step retrieval-then-ranking pipeline. However, our statement refers specifically to the **training cost** of directly optimizing Hits@K-style objectives, which require full pairwise rankings and are thus infeasible at scale.
>
> The distinction between training and inference is crucial here. While SEAL, BUDDY, and similar models use a two-step retrieval-then-ranking strategy during inference, they are not trained using candidate sets. Rather, **they are trained via binary classification over uniformly sampled connected and disconnected node pairs, which is precisely the set up of the standard link prediction benchmark**.
>
> While one may consider training a ranking model using a pre-trained retriever, this introduces a dependency on the retriever. If the retriever fails to retrieve true neighbors, then even a strong ranking model cannot recover. This is why we believe evaluating retrievers directly on recommendation metrics is essential; retrieval is the first step in the pipeline, and its quality bounds the overall system performance.
>
> We appreciate the reviewer's thoughtful feedback, which prompted us to clarify this key distinction. We hope this response strengthens the overall clarity of the paper.

---

> > ### Comment · Reviewer_vU2g · 2025-04-03
> >
> > I appreciate the authors' further explanation and plan to update the manuscript accordingly. I'll raise my score.

---

### Decision · Program_Chairs · 2025-05-01

**Decision:**

Accept (poster)

**Comment:**

This paper addresses link prediction problem an argues that a common link sampling procedure used in evaluations is prone to produce biased results toward high degree nodes. The authors analyze this problem both empirically and theoretically. They also propose a degree-corrected link prediction benchmark and show that it provides a more robust evaluation framework for link prediction task. The reviewers raised several issues, including  the evaluation metric, which the authors addressed in their rebuttal. Overall, the consensus is that the paper presents a solid and interesting contribution and should be accepted to the conference.